



# 1 Real-Time Pollen Identification using Holographic Imaging and
# 2 Fluorescence Measurement

Sophie Erb[1,2,*], Elias Graf[3,*], Yanick Zeder[3,*], Simone Lionetti[4], Alexis Berne[2], Bernard Clot[1], Gian
Lieberherr[1], Fiona Tummon[1], Pascal Wullschleger[4], Benoît Crouzy[1]
[*]These authors contributed equally to this work.
[1] Federal Office of Meteorology and Climatology MeteoSwiss, Chemin de l'Aérologie, CH-1530, Payerne, Switzerland
[2] Environmental Remote Sensing Laboratory (LTE), École Polytechnique Fédérale de Lausanne, Lausanne, Switzerland
[3] Swisens AG, Horw, CH-6048, Switzerland
[4]Algorithmic Business Research Lab (ABIZ), Lucerne University of Applied Sciences and Arts, Lucerne, Switzerland
*Correspondence to*: Sophie Erb (sophie.erb@meteoswiss.ch)
**Abstract.** Over the past few years, a diverse range of automatic real-time instruments has been developed to respond to the
needs of end users in terms of information about atmospheric bioaerosols. One of them, the SwisensPoleno Jupiter, is an
airflow cytometer used for operational automatic bioaerosol monitoring. The instrument records holographic images and
fluorescence information for single aerosol particles, which can be used for identification of several aerosol types, in particular
different pollen taxa. To improve the pollen identification algorithm applied to the SwisensPoleno Jupiter and currently based
only on the holography data, we explore the impact of merging fluorescence spectra measurements with holographic images.
We demonstrate that combining information from these two sources results in a considerable improvement in the classification
performance compared to using only a single source (balanced accuracy of 0.992 vs. 0.968 and 0.878). This increase in
performance can be ascribed to the fact that often classes which are difficult to resolve using holography alone can be well
identified using fluorescence and vice versa. We also present a detailed statistical analysis of the features of the pollen grains
that are measured and provide a robust, physically-based insight into the algorithm's identification process. The results are
expected to have a direct impact on operational pollen identification models, particularly improving the recognition of taxa
responsible for respiratory allergies.

## 25    1.    Introduction

Over the past decades a considerable increase in aeroallergen-related diseases such as asthma or allergic rhinitis has been
observed (Ring et al. 2001; Woolcock et al. 2001; Woolcock et Peat 2007). This has resulted in a rise in associated direct and
indirect health costs in terms of hospitalisation, medication costs and absence from work (Zuberbier et al. 2014; Greiner et al.
2011). Currently, the prevalence of pollen allergy ranges between 10 to 30% of the population in Westernised countries and
up to 40% of children in high-income countries (Pawankar et al. 2011). In future, the relevance of pollen as an allergen may
increase further as a result of climate change, which perturbs the life cycle of plants through drier environmental conditions
and increased temperatures. Stressed plants tend to have an earlier and/or longer blooming season (Ziello et al. 2012) and



produce more pollen with higher concentrations of allergens (Damialis et al. 2019; Beggs 2016; D'Amato et al. 2016), possibly
further contributing to the increase and severity of allergic diseases. For these reasons, systems to measure airborne pollen
concentrations are essential to meet public health challenges associated with respiratory allergies. Through real-time
measurements and the development of forecast models (Chappuis et al. 2020), they can help reduce health costs with better
diagnosis and prevention, thus helping patients to better manage their symptoms.

Most European countries started monitoring pollen in the second half of the 20[th] century using Hirst-type instruments (Hirst
1952) with manual identification and counting part of the process (Clot 2003; Spieksma 1990). However, this method provides
data at low time resolution, typically daily mean values, after a processing time of up to 10 days. The spread of pollen grains
on the collection band and the limited sampling (Oteros et al. 2017) mean that data at higher temporal resolutions, or at low
concentrations (below 10 pollen grains/m$^3$), have considerably increased uncertainty (Adamov et al. 2021). Although little
data is available to study atmospheric pollen phenomena at high temporal resolutions, it is widely expected that pollen
production and dispersal processes take place at sub-daily scales since they are highly influenced by local meteorological
environmental conditions (Rojo et al. 2015; Rantio-Lehtimäki 1994). Provision of real-time pollen data is also crucial for
forecasting purposes, since models can then integrate these real-time data to deliver considerably improved forecasts (Sofiev

48     2019).


Over the past few years, several instruments designed for real-time pollen monitoring have come onto the market (Crouzy et
al. 2016; Oteros et al. 2015), as comprehensively reviewed in previous work (Huffman et al. 2020; Buters et al. 2022; Maya
Manzano et al. 2023). Among the most promising instruments are airflow cytometers which allow the characterisation of
particles almost in real-time as they pass through the instrument and enable continuous monitoring with high temporal
resolution (10 minutes as for weather parameters or below) over a whole season. In particular, the SwisensPoleno Jupiter
(developed by Swisens AG, Switzerland) is an instrument for bioaerosol identification which can take in-flight holographic
images of particles and measure their fluorescence (FL hereafter) (Sauvageat et al. 2020; Tummon et al. 2021; Lieberherr et
al. 2021). Coupled with a machine learning (ML) algorithm, it has been shown to perform well for pollen monitoring even if
the algorithm uses just the holographic data (Sauvageat et al. 2020; Crouzy et al. 2022; Maya Manzano et al. 2023).

The FL data has to date not been used for pollen identification with the SwisensPoleno Jupiter. Sauvageat et al. 2020 reached
an accuracy above 96% for the main allergenic pollen species (*Ambrosia artemisiifolia*, *Corylus avellana*, *Dactylis glomerata*,
*Fagus sylvatica*, *Fraxinus excelsior*, *Pinus sylvestris*, *Quercus robur* and *Urtica dioica*) using only holographic images.
However, some species have similar morphologies which can cause misclassifications and thus lower the algorithm
performance, as previously identified in Sauvageat et al. 2020. In this paper, we investigate whether FL helps discriminate
single pollen grains between different allergenic taxa based on their chemical compositions to reduce the level of confusion



resulting from their similar shapes. Moreover, we also verify whether the FL measurements are consistent for each species
when using different SwisensPoleno units.

## 2.  Material and Methods

In this work we investigate the impact of including the set of FL measurements, constituting the particle FL spectra, as input
for pollen identification using artificial neural networks. We trained and assessed the performance of three neural networks
with the same dataset but using different inputs: only holographic images (holo), only FL spectra (FL), or both (combined).
The performance of each model is evaluated using classical metrics, here the balanced accuracy, the F1-score, and Matthew's
Correlation Coefficient (MCC) as defined in Chicco et al. 2020, as well as the (relative) error rate derived from the accuracy.

### 2.1.  Pollen holography and fluorescence dataset

The SwisensPoleno Jupiter measures particles in flight as they pass through the instrument. When a particle triggers the
detector, holographic images are taken by two cameras which are both orthogonal to the direction of flight and at 90° to each
other. These images are greyscale with a resolution of 200 by 200 pixels after numerical reconstruction and cropping, with
each pixel representing a square of 0.595 x 0.595 $\mu$m in the physical domain. FL is then sequentially induced by three excitation
sources and captured in five different wavelength channels, for a total of 15 measured FL intensities. The FL lifetime is also
measured but is not used in the present work. The three different excitation wavelengths are 405, 365, and 280 nm, while the
reception wavebands are 333-381, 411-459, 465-501, 539-585, and 658-694 nm. In the following, we will refer to each
waveband by its central wavelength, i.e., 357, 435, 483, 562, and 676 nm. Note that the first measurement channel is saturated
by scattered light when the 365 nm excitation source is activated. Also, for single-photon excitation, we expect to measure no
signal in the first measurement channel when the 405 nm source is active. This effectively reduces the useful intensity
measurements to 13. The FL data requires additional pre-processing to simplify its usability and improve robustness. More
details on these steps are provided in Section 2.2. Finally, the SwisensPoleno Jupiter also performs polarised scattered light
measurements, which are however not used in the present work. We therefore limit the analysis to characterisation of particle
morphology using digital holography and chemical composition with FL intensity measurements. From hereon, we refer to
the set of holographic images and FL measurements for each individual particle as "an event". A more extensive description
of the data collection process is provided in Sauvageat et al. 2020.

This study is based on a pollen dataset created by aerosolising freshly collected pollen at the Swiss Federal Office of
Meteorology and Climatology MeteoSwiss (hereafter MeteoSwiss) station in Payerne, Switzerland. In total, the dataset
consists of measurements from 57'300 pollen grains distributed among seven different wind-pollinated and allergy relevant
plant taxa as reported in Table 1. For simplicity, we will refer to these taxa also as "classes" and only the genus name will be
used to refer to each of them. In Figure 1, we present examples of reconstructed images for the different classes considered in
this work. To compare results across different instruments (of the same type), all measurements were performed using two





SwisensPoleno Jupiter systems denoted P4 and P5. The counts for each pollen taxa and SwisensPoleno are also given in Table

99 1.


The pollen samples were collected from a single tree for *Alnus*, *Betula*, *Corylus*, *Fagus,* and *Quercus*, from two different trees
for *Fraxinus,* and from a few neighbouring stems for the grass *Cynosurus*. After outdoor collection, pollen was brought to the
measurement site and aerosolised. This was achieved using a SwisensAtomizer which disperses particles using a vibrating
membrane and an airstream. Samples are thus scattered in a chamber and drawn into the instrument, producing a regular flow
of pollen grains. To prevent the pollen from drying out, plants that were not more than 15 km away from the MeteoSwiss
station were selected, which means it was possible to aerosolise samples soon after collection (usually within one hour). Pollen
samples were analysed using two instruments one after another implying a time lag between the data for P4 and P5, which
ranges from just 35 minutes for *Alnus* to 80 minutes for *Quercus* (the mean time lag is 60 minutes). For *Fraxinus* there is no
such lag since the data come from two different samples that were measured on different days. Datasets for all the considered
pollen taxa were created in 2020, except *Alnus* and *Corylus* which are from early 2021.

| Common name | Latin scientific name | Number of events for P4 | Number of events for P5 |
|---|---|---|---|
| Alder | *Alnus glutinosa* | 8416 | 2643 |
| Birch | *Betula pendula* | 6128 | 5458 |
| Hazel | *Corylus avellana* | 4714 | 4444 |
| Crested Dog's-Tail (Grass) | *Cynosurus cristatus* | 5895 | 2117 |
| Beech | *Fagus sylvatica* | 2178 | 2827 |
| Ash | *Fraxinus excelsior* | 2557 | 4837 |
| Oak | *Quercus robur* | 3036 | 2050 |
| | TOTAL | 32924 | 24376 |

**Table 1: Distribution of pollen counts per taxa and Poleno.**

**2.2.    Data pre-processing**
The datasets required to train the algorithms were generated as follows. First, the holographic data for each class were cleaned
to eliminate any non-pollen events or events associated with other pollen taxa. This was achieved with additional filters on
shape properties (image features computed after binarisation as described in Sauvageat et al. 2020), which were appropriately
selected for every class by heuristic visual inspection of the holographic images. Thereafter, for each event the background
signal caused by scattered light was subtracted from the raw FL measurement. This background especially disturbs the low FL

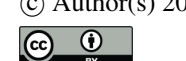

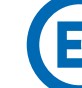

intensity measurements where the scattered light dominates relative to the particle signal. The background signal was obtained
by conducting measurements with no particles present in the measurement chamber, leaving just the scattered light induced
by the excitation source. If the subtraction caused the final signal to be negative due to noise, the resulting value was set to
zero. Finally, since the absolute FL compensated by the scattered light is still dependent on the measuring system, the particle
size, and the particle position within the measurement volume, we transformed it into relative FL. Namely, the relative
fluorescence intensity $r_{ij}$ for measurement channel $i$ and excitation source $j$ is obtained by dividing the absolute FL intensity
$a_{ij}$ by the sum of the FL intensities on all channels $k$ for the same excitation source $j$ :
$$r_{ij} = \frac{a_{ij}}{\sum_k a_{kj}}$$

Using relative FL, although we lose the absolute FL intensities, allows measurement systems to be compared without specific
data modification. The inter-compatibility aspect is especially important when considering a measurement network. Thanks to
this standardisation, the same algorithm can be used for all systems in the network rather than adjusting the classification
algorithm individually for each measurement system.
### 2.3.    Data exploration
Before applying any ML algorithm, it is important to explore the data to better understand their characteristics. In the following,
the distributions of the various holographic image features as well as typical relative FL spectra for the different pollen types
are investigated. We also explore the structure of the data using dimension reduction.

To get other characteristic features from the reconstructed holographic images, further image processing steps are conducted
using the Python package "Scikit" (Van der Walt et al. 2014). Physically-based particle features, such as the minor and major
axes, the area, the eccentricity, and the particle brightness (mean intensity of the pixels reproducing the particle) are computed
for each image separately. Other statistics were calculated based on image features, e.g., the equivalent area diameter defined
as the diameter of a circle with the same area as the particle. The distribution of these features for each pollen class and each
measurement system were analysed separately and are presented in the Results section.

As previously discussed, alongside the holography images, relative FL spectra are used for enhanced characterisation of the
pollen grains. During data exploration, we observed inconsistent results for the 405 nm laser excitation, which upon further
inspection revealed a misalignment of this laser in one of the measurement systems. For this reason, we will only use the 280
nm and 365 nm excitation throughout the rest of the present work. The distributions of the valid FL spectra are presented and
discussed in the Results section.

As a way to explore all the features of the dataset at once, we performed dimensionality reduction. We used the Uniform
Manifold Approximation and Projection (McInnes et al. 2020), called UMAP, on the input data of each model (holo, FL and



combined). This technique allows us to plot multidimensional data as points on a plane, therefore it gives an insight on how
similar/different data points are depending on how far from another they are in the plane.

### 2.4.    Machine Learning Model

To handle the classification task, we randomly split the data into training (75%) and test (25%) sets and chose a multi-layer
"deep" artificial neural network to learn how to identify pollen grains based on the training set. This network maps input data
from the holographic images and relative FL spectra to the different pollen classes. The full network, built using the ML
framework Keras (Chollet et al. 2015), is shown in Figure 2. To handle the image input, an EfficientNet B0 model pre-trained
on ImageNet is used (Tan et Le 2019). It achieves state-of-the-art performance for classification tasks. For treating the spectral
information, a single fully connected (denoted FC hereafter) hidden layer with 255 neurons is used. As a pre-trained model,
the parameters of EfficientNet B0 are frozen and therefore not modified in the training on the pollen dataset. However, the
parameters of the layers after it are optimised according to the training data. The results of the two feature extraction networks
are concatenated, then dropout is added and finally the result is passed to the decision layer. The width of this FC decision
layer matches the number of classes (seven in this case). Lastly, the output is normalised by a softmax layer to obtain a
probability distribution. To compute the loss, we used the cross-entropy function between the predicted and reference classes.
To ensure a fair comparison, each model was trained for exactly 200 epochs. In training runs where only images or only relative
FL spectra were used, the path not used was removed from the model graph (Figure 2). The figure shows the model with both
features active.
The models were evaluated using a test set consisting of 25% of the data from both instruments, sampled randomly. We used
balanced accuracy, F1 score and Matthew's Correlation Coefficient (MCC) as metrics to assess the model performance. For
accuracy, the corresponding confidence intervals were calculated via normal approximation as explained in Raschka 2020. It
is important to note that the model used here is a baseline and has not undergone hyper-parameter optimisation, therefore no
validation set has been defined in order to keep a maximum of data for training. This means that a degradation of scores is
possible when applying the model to operational data. Nonetheless, the present study does not aim to provide an operational
model but simply investigate the potential of using FL as a complement to holography for single particle identification.

## 3.    Results

### 3.1.    Feature observations

Important observations can already be made by looking at basic geometrical features derived from holographic images. As an
example, we consider the distributions of equivalent area diameter and eccentricity in Figure 3 (a) and (b). Note that for
geometrical features, the value associated with each particle is the largest result obtained for the pair of holographic images.
Regarding the equivalent area diameter, its distribution provides information about the size of the pollen grains for a given
class. As illustrated in Figure 3 (a), *Fagus* pollen grains are typically large with a maximum equivalent area diameter of 45-55





μm, which corresponds to the literature (Halbritter et al. 2021) and is clearly superior to all other classes we considered in our
study. Conversely, the distribution of the eccentricity gives an insight on how round the pollen grains are. In that case,
*Cynosurus* pollen grains have the roundest shape with a maximal eccentricity between 0.4 and 0.55 (0 representing a circle
and 1 an ellipse), whereas *Quercus*' values are in the range 0.8-0.9 due to its more elliptical shape. These characteristics can
also be observed on the holographic images in Figure 1.

The distributions of the relative FL spectra allow us to identify some classes that have distinct FL signatures. Figure 3 (c) and
(d) show the distribution of the relative FL for the two excitation-emission combinations where the differences between taxa
are the largest. The excitation sources are at 280 and 365 nm with emission channels at 357 and 435 nm respectively. In Figure
3, we observe, for both plot (c) and (d), clear differences in relative FL for *Cynosurus*, which presents considerably higher
values compared to the other taxa. In addition, differences between instruments show that P4 and P5 have similar
measurements in the 280/357 nm but P5 has significantly lower measurements for *Corylus* and *Cynosurus* in the 365/435 nm.
Overall, all combinations of excitation sources and emission channels provide relevant information for pollen characterisation
and the ones presented in Figure 3 (c) and (d) represent well the type of patterns that can be observed.

Finally, the UMAP plots, given in the left column of Figure 4, show how different or similar are the image and FL features of
each taxon. We observe a clear distinction based on morphology (Figure 4 (a)) for *Fagus* and *Quercus*, with *Cynosurus* also
having only little overlap with *Corylus*. However, the latter and especially *Betula* and *Alnus* are clearly mixed up. In Figure 4
(b), the UMAP on FL spectra does not exhibit the same group structure as for morphology. Here, *Fagus* and *Cynosurus* are
plainly detached from the remaining groups which are themselves imbricated. Ultimately, all groups are fully separated when
building the UMAP on both morphology and FL features. We observe a correspondence between the separation of groups on
the UMAPs and the capacity of the ML model to classify those classes correctly.
**3.2.  Classification performance**
The classification results for each model are given as confusion matrices in Figure 4 and summarised in Table 2. We observe
in these results that the holo model globally performs better than the FL model when training on a single modality. The FL
model indeed encounters difficulties distinguishing some classes such as *Quercus* and *Fraxinus* or *Betula* and *Corylus* (Figure
4 (b)) which exhibit similar relative FL spectra. When considering the morphology of *Quercus* and *Fraxinus* (Figure 3 (a) and
(b)), it is not surprising that the holography model performs better at differentiating these classes as they present significantly
distinct shapes. As the performance for the single-input models here is already (very) high, minor dips in performance can
make a notable difference. Combining holography and FL improves the performance compared to the single input models for
every taxon considered, except for *Fagus* and *Cynosurus* that already obtain perfect scores with single input models. The
performance gain is noteworthy as the combined model achieves an overall balanced accuracy of 99.2% compared to either
96.8% or 87.8% for the individual holography or FL models respectively. As a complement, the confidence intervals associated



with the accuracy of each model for each taxon are displayed in Figure 5. The non-overlapping of the confidence intervals
indicates a statistical difference between accuracies. The combined model outperforms both single-input models for five of the
seven taxa, namely, *Alnus*, *Betula*, *Corylus, Fraxinus* and *Quercus*. Thus, logically, the balanced accuracies of the holo and
FL models are significantly lower than that of the combined model (see Table 2). It follows that the absolute error rates, defined
as 1 minus the accuracy, of the holo- (3.2%) and FL-only (12.2%) models are respectively 4 and 15 times higher than that of
the combined model (0.8%). This indicates that mistakes in particle identification occur for roughly 3 particles over 100 for
the holo model, 12 particles over 100 for the FL model but less than 1 particle over 100 for the combined model.

| Model | Balanced accuracy | F1-score | MCC |
|---|---|---|---|
| Holography only | 0.968, [0.965; 0.970] | 0.964 | 0.958 |
| FL only | 0.878, [0.874; 0.882] | 0.890 | 0.874 |
| Combined | 0.992, [0.991; 0.993] | 0.992 | 0.991 |

**Table 2: Classification performance of each model. The balanced accuracy, with its associated 95% confidence interval,**
**represents the average of the recalls (ratio of correct prediction over total events for each class), ranging from 0 to 1.**
**The F1-score is the harmonic mean of the precision and recall, ranging from 0 to 1 and MCC stands for Matthew's**
**Correlation Coefficient and is a robust metric for classification performance, ranging from -1 to 1.**
**4.    Discussion**
The results show that combining FL with holography leads to a substantial identification performance gain. The differences
between the combined model accuracy and both single-input models confirm the findings from the UMAPs. This demonstrates
that by combining the two inputs, the complementary morphological and biochemical properties of pollen grains can be used
for a better classification. Although it seems small, the gain in accuracy is important for the field of aerobiology and specifically
pollen monitoring since pollen grains only represent a minor part of all the particles in the air. Since pollen concentrations
typically range from a few grains (< 10) to a few hundred grains per cubic metre, and the thresholds for allergy symptoms are
usually around a few tens of grains per cubic metre (Gehrig Bichsel et al. 2017; Pollen.lu 2003), misclassifications can have
an impact on the information provided to allergic people. Above all, high identification accuracy is particularly important for
plants with highly allergenic pollen such as *Ambrosia artemisiifolia* (common ragweed) as a few grains are sufficient to cause
allergy symptoms.

Not only is the combined model's accuracy superior to the other models, but this gain is specifically important for some key
pollen taxa. Indeed, the group of *Alnus*, *Betula* and *Corylus*, all from the Betulaceae family, is known to be difficult to classify
accurately and presents a very high allergic potency with possible cross-reactivity in central and northern Europe (Puc et
Kasprzyk 2013). Thus, the excellent classification performance obtained here opens the gate towards better monitoring by
using holography together with fluorescence data. In addition, the consistent FL signal in between instruments and the available



excitation sources and measurement channels characterise single pollen grains precisely even though the 405 nm excitation
source was set aside. Also, the combinations of excitation and emission wavelengths used in the Poleno correspond to the most
prominent fluorescence modes for a variety of dry pollen studied in (Pöhlker et al. 2013). Then, the coherence of the
fluorescence spectra obtained here with the measurements of the latter study, brings confidence into our measurements and
the measurement instrument per se. In future work, the 405 nm excitation source needs to be included to verify its potential
for improvement.
When working with images, choosing neural networks for classification is the obvious solution to be sure not to lose
information by using the image itself as input. However, the discrimination of pollen taxa using the UMAP dimension
reduction method shows that working with features derived from the holographic images is also a possibility for pollen
classification. Future work testing other machine learning methods on image features and fluorescence spectra needs to be
conducted as other classifiers may perform similarly while being cheaper in terms of computational resources.
In the end, we expect the benefit of combining holography with FL measurements for pollen classification to have a positive
impact on the capacity of models to discriminate different pollen taxa. Moreover, in an operational setup, the benefit of using
FL in addition to holography could be even higher as it would allow for an easy distinction between biological and non-
biological particles (e.g. water droplets, sand particles or dust) assuming that they do not fluoresce. Yet, the extent of the gain
in the real case scenario remains to be quantified as the dataset used in this study probably does not catch all the environmental
variability.
**5.    Conclusion**
The present study demonstrates the benefit of using FL measurements as a complementary input to holographic images for
single-grain pollen identification using the SwisensPoleno and ML algorithms for the most important allergy causing pollen
taxa in Central Europe. The capacity of the ML model to identify pollen grains depends on both inputs and they compensate
each other when one does not provide enough information for accurate identification. As a result, the performance of the
combined model is systematically higher than either of the models trained with a single input. The restricted and manually
created dataset used in this study has several limitations, but it still provides strong evidence for the complementary role of FL
and holography.
In conclusion, we recommend the use of relative FL as a secondary input for automatic pollen identification using the
SwisensPoleno Jupiter. In this study, we tested its contribution on a restricted dataset, showing that the contribution of FL is
of great value for operational networks where similar pollen taxa can be encountered. Finally, the use of relative FL for
automatic pollen identification further opens the door towards a larger and more precise monitoring of bioaerosols. For



example, objects which are challenging to identify using holographic imaging only, such as fungal spores, could be added to
the panel of particles.
**Author contribution**
EG, SE and YZ conducted the study and contributed equally as main authors. SL guided the machine learning aspects and
supervised PW in his work on the relative fluorescence. AB, BCl, GL and FT contributed to writing, and BCr supervised the
study and contributed to writing.
**Competing interests**
EG and YZ are employees of Swisens AG, and AB is a member of the editorial board of AMT. The investigations were carried
out in compliance with good scientific practices and the declared relationships have no effect on the results presented. The
peer-review process was guided by an independent editor, and the authors have also no other competing interests to declare.

**Acknowledgements**
We would like to thank all the co-authors for their support, advice and help in the various aspects of this study. This work was
funded by the Swiss National Science Foundation (IZCOZ0_198117).



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



**Figures**

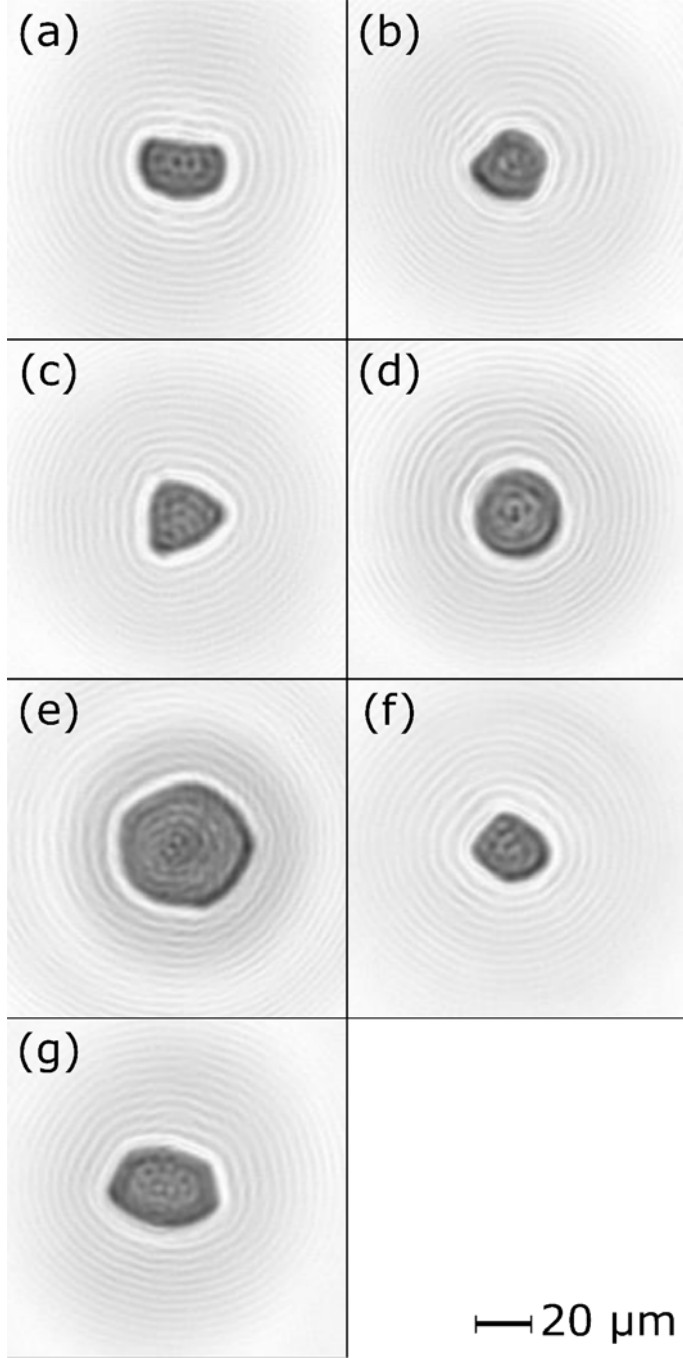


**Figure 1: Holographic images of pollen after numerical reconstruction: (a)** *Alnus glutinosa* **(b)** *Betula pendula***, (c)**
*Corylus avellana***, (d)** *Cynosurus cristatus***, (e)** *Fagus sylvatica***, (f)** *Fraxinus excelsior***, (g)** *Quercus robur*





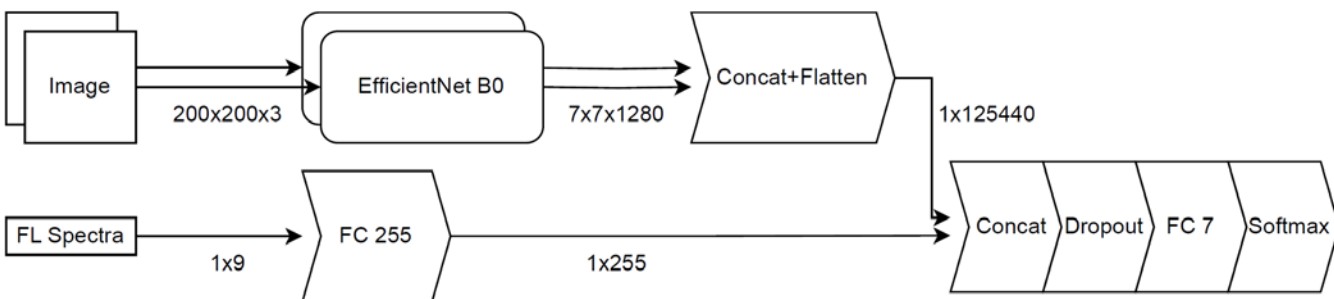

**Figure 2: ML model structure used to classify the pollen data. The top path handles the holographic image data while the bottom path processes the relative FL spectra data. The numbers on the connecting lines denote the dimensions of the data.**





**Figure 3: Distribution of holographic image features (upper plots) and relative FL (bottom plots) for each pollen class and measurement system. (a) Maximum equivalent area diameter in $\mu$m, defined as the diameter of a circle with the same area as the particle, (b) Maximum eccentricity, defined as the deviation of the ellipse fitted to the particle from a perfect circle, ranging from 0 for a circle to close to 1 for an ellipse. (c) Measured relative FL intensity with 280 nm excitation and detector with centre wavelength 357 nm and (d) with 365 nm excitation and detector 562 nm.**



**Figure 4: Left side: Uniform Manifold Approximation and Projection (UMAP) of event features (morphology or/and FL features). Right side: Confusion matrices indicating the performance of each model on the test set. Line (a) holography only, line (b) relative FL only and line (c) combined relative FL and holography. UMAP settings: neighbours = 15, minimum distance = 0.001, random state = 42.**





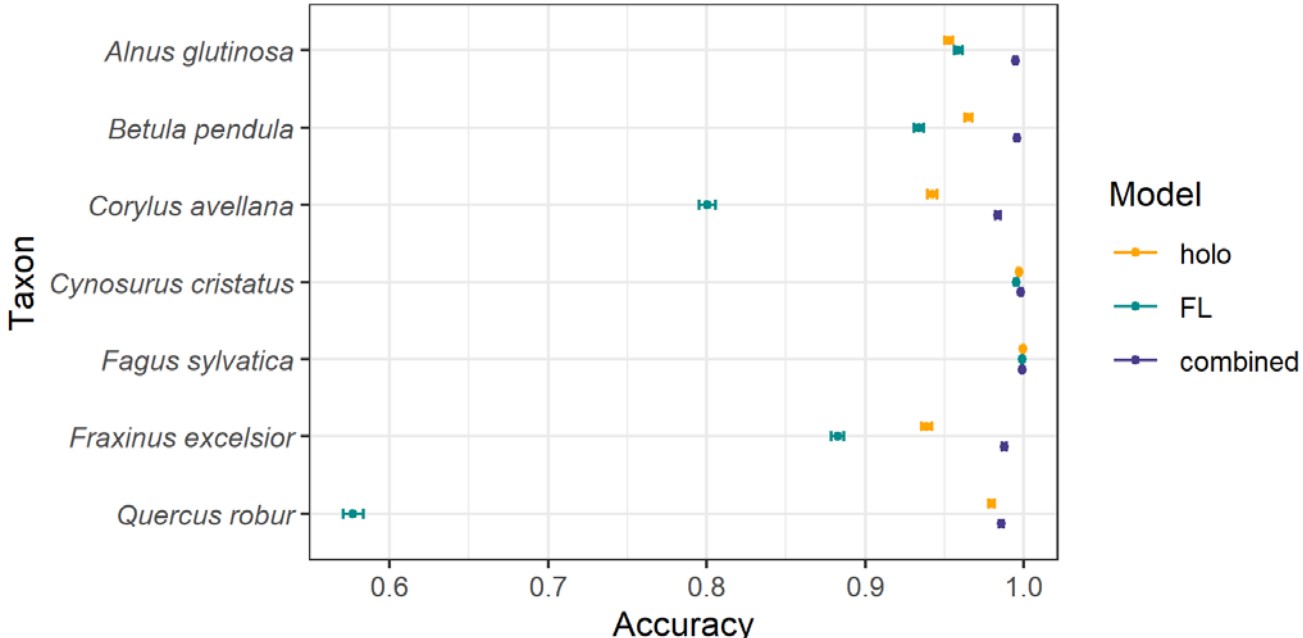

Figure 5: Accuracy of each model for each taxon. The error bars represent the 95% confidence intervals.