# Peer review of "Real-Time Pollen Identification using Holographic Imaging and Fluorescence Measurement"

_EGUsphere, 2023_

## Author Comment (AC1)

**RC1**: Anonymous Referee #1, 04 Sep 2023
**General comments**

This paper describes a new measurement technique for identifying pollen grains that combines holography and fluorescence. Overall, the paper reports methods and laboratory results that will be useful to the pollen observation community, although with the caveat that this has only been tested and trained in the lab and has not yet been used on ambient samples. It would be helpful to include something to this effect (e.g, laboratory only, needs to be tested in ambient air) in the abstract, as it only is raised briefly in the discussion. I would have preferred to see some ambient samples presented in the paper but recognize that this increases the scope of work substantially. Otherwise, the technique holds promise for distinguishing different types of pollen for real-time sampling, which is an exciting result. I have several minor presentation comments below that would make this manuscript acceptable for publication.

We thank the referee for their positive feedback on the manuscript. We agree on the importance of clarifying the scope of the study. The measurements were performed outdoor and not in the laboratory as suggested by the Referee. To this effect, material collected on neighboring plants was used. We have added the following comments to the revised version of the manuscript to clarify this point:

L18: Added ", using manually generated data," to underline that we do not use operational data.

L105-106: "After collection, pollen was brought to the outdoor measurement site and aerosolised.", to precise the outdoor measurements.

L232-233: "The results based on manually generated data show that combining FL with holography leads to a substantial identification performance gain.", to underline the manually generated data.

L270-272: "The present study demonstrates the potential of using FL measurements as a complementary input to holographic images for single-grain pollen identification using the SwisensPoleno and ML algorithms for the most important allergy causing pollen taxa in Central Europe.", changed benefit to potential to nuance the message.

Below we address the presentation comments by the Referee in the order as they appear.

**Minor comments**

1. Title: "measurement" -> "measurements"

   Done

2. Line 61: "... for the main allergenic species...." – is this the main species in Switzerland, or more broadly in Europe? More clarity for the selection of these seven pollen types would be beneficial.

   It is true for Switzerland but also representative for central Europe. Grass pollen (here *Dactylis glomerata*) is the most impacting pollen type in central Europe. Ragweed (*Ambrosia artemisiifolia*) is the most allergenic and the other five (*Corylus avellana, Fagus sylvatica, Fraxinus excelsior, Pinus sylvestris, Quercus robur* and *Urtica dioica*) are amongst the most common pollen types in this region. This list should also be completed by Birch (*Betula*

*pendula*) but no sample was available at the time of the study. We have added the following comment to clarify this point:

L61: precision "… for eight of the main allergenic pollen species in central Europe …"

3. Line 75: What size of particles trigger the detector? While this study is specifying the types of pollen evaluated, what would happen if it were an ambient air sample?

The detector triggers particles of a minimal size of 0.5 µm, however below 2 µm there is very limited information for holography due to the resolution of 0.595 µm/pixel. If the model was applied to ambient air samples (operational data), we would filter the data before giving it to the model. It is usually done this way because Neural Networks are a supervised classification method, meaning they can classify correctly only what they learned to distinguish. Usually for pollen, a simple filter on the particle aera and solidity is sufficient to remove debris and smaller particles. The general concept of prefiltering before classifying is described in Sauvageat et al. 2020.

L77: added "…, in the size range from 0.5 to 300 µm, …"

L176-178: "This means that a degradation of scores is possible when applying the model to operational data as all sorts of pollen taxa can be encountered considering that other particles are filtered out before the classification.", to precise the process in case of operational data

4. There is some terminology in the paper that is rather confusing:
   1. "event" – this is defined on line 89, but that makes it sound like you are sampling ambient air versus a controlled emission. Also confusing in Table 1, where it really seems to be the number of particles counted and evaluated. I would suggest something like "number of particles counted", "number of images", or even just "pollen count".

The term "event" represents the input data to a model. We defined the term "event" for simplicity as it represents the data recorded for each particle measured by the instrument independently of the nature of the data. This way, we do not have to specify that we have "holographic images and fluorescence spectra" each time we speak about a measured particle. An event should be seen as an occurrence of the measurement of a particle and does correspond to the number of pollen grains that were measured by the SwisensPoleno. We used the term "event" as it also corresponds to the terminology of the SwisensPoleno. For consistency with previous work, we prefer to keep that term. However, we checked that its use is clear along the paper and made a few corrections in that regard:

L225: Caption of Table 2: "… ratio of correct prediction over total count for each class… "

L407: Caption of Figure 4: "Uniform Manifold Approximation and Projection (UMAP) of particle features (morphology or/and FL features) …"

   2. "class" – defined on line 95 as the same as plant taxa, although it was unclear why this specific term was used – why not just keep "taxa"? Also, later throughout the paper (e.g., line 218, 241, y-axis label on Figure 5) these are used interchangeably and makes it rather confusing.

The term "class" comes from the Machine Learning field. Using the word class instead of taxa makes the paper less specific to pollen as other biological or non-biological aerosols can be measured using the same instrument and method.

5. Table 1: If using the term "class" in the paper, I would suggest changing the header on the first column from "Common name" to "Class (common name)" to clarify the terms more clearly.

Change done as suggested.

6. Line 179: eccentricity seems to be a useful metric, but is there also one about symmetry that might be helpful for more complex grains?

We thank the Referee for the useful suggestion. We have added the following comment to the revised version of the manuscript.

"While the eccentricity is used to give a hint on the symmetry of the pollen grain, further metrics could be introduced to further quantify symmetry. This was not implemented in the present study as feature extraction is done automatically by the convolutional neural network."

7. In Figure 3c, the standard deviation on the relative fluorescence is extremely large. Can the authors comment on why they think that this metric improves the ML model?

In Figure 3c, we displayed the relative fluorescence measured when exciting the particle with a 280nm laser and receiving on emission channel at 357nm. This is one metric among the complete spectrum of the particle. Even though the variation on this metric is large, it still gives useful information when considering the complete set of intensities (13 in total). When considering the spectrum, we have a combination of values, and this is the full pattern that is given to the model.

8. Line 183: "superior" -> "larger"

L183: replaced "superior to" by "larger than"

9. Line 247-248: Please rephrase

L247-248: rephrased as "The coherence between our results and those from Pöhlker et al. 2013 brings confidence into our measurements and the stability of the Poleno."

We thank the referee for their time and implication in our work. The comments were constructive and helped us improve the quality of the paper.

---

## Author Comment (AC2)

**RC2**: Anonymous Referee #2, 16 Oct 2023

This manuscript evaluates the effectiveness of including fluorescence information along with holographic images for the identification of a small number of allergenic pollen species. They find that the accuracy of the combined technique is a large improvement over the use of only fluorescence or only holography for identification. While this is a nice piece of work that shows value in this combination, I do think it is quite limited in applicability due to the small number of pollen samples examined. A few questions/issues that I hope the authors can address are as follows:

I would recommend a bit more instrument description early in the paper. I realize that the instrument has been described previously but it would be helpful to have at least a brief summary of the most pertinent details presented here since it is not an instrument that has been written about extensively and may be unfamiliar to some readers. It would be nice to know the detectable size range, what the sizing laser and the fluorescence excitation sources are, how the excitation is triggered by the instrument and how the scattered/emitted light is collected.

We agree with the referee on the need to make the paper more self-contained. We added the suggested content as:

L75: detectable size range "The SwisensPoleno Jupiter measures particles in flight, in the size range from 0.5 to 300µm, as they pass through the instrument."

L80: how the excitation is triggered by the instrument and how the scattered/emitted light is collected, we added "Right after the particle triggering and holographic images, FL is measured using the Laser Induced Fluorescence (LIF) method. FL is then sequentially induced by three excitation sources and captured in five different wavelength channels, for a total of 15 measured FL intensities. For each source, the FL is induced by shooting at the particle as it passes the detector and the FL subsequently emitted by the particle is captured by Silicon Photomultipliers (SiPM)."

L80-81: excitation sources already given "The three different excitation wavelengths are 405, 365, and 280 nm, while the reception wavebands are 333-381, 411-459, 465-501, 539-585, and 658-694 nm."

Along similar lines, I think it would be good to describe the data pre-processing in section 2.2 a little more clearly. I believe you mean that a human user removed both non-pollen particles and any pollen particles that were not of the particular class being sampled. If I'm interpreting that correctly, that seems like quite a limitation. If the plan is to eventually use this technique on ambient data the instrument and the ML methodology is going to need to be able to deal with both non-pollen particles and sampling periods in which many different kinds of pollen are mixed together. If this cleaning were not done, how much worse would the performance be? Can you comment on whether this would be a necessary step in processing ambient data and, if so, how that might be accomplished and what it would mean for identification of different pollen types?

First, the rationale for cleaning the data is that we have data that we generated ourselves (we atomised pollen and fed the instrument with it), thus we may have particles that we would not find in ambient air (for example pollen aggregates), and this could bias the model when learning form these datasets. Secondly, we use a 2-step process when applying the model to ambient air data. A) We apply a "pollen filter" based on the particle aera and solidity (removing very small particles and those with irregular shapes since pollen grains typically have smooth convex shapes) and B) we apply the ML model to the remaining data. With this process, the ML model is able to differentiate

various taxa well. In short, the approach is identical to what is done for operational automatic pollen monitoring based on digital holography only.

The authors state that, if subtraction of the baseline resulted in apparently negative fluorescence values, those values were assumed to be zero instead. Although this is likely a minor effect, I think it is more mathematically correct to include negative values as, without them, you are necessarily biasing the data high. Aren't they statistically meaningful in the sense that they reflect baseline variability in the instrument?

They indeed reflect a certain variability due to noise, but we did not keep them because during the computation of ratios, for very low fluorescence particles, they may compensate positive values and lead to division by zero. This generates numerical instabilities, which our machine learning models cannot deal with, while they otherwise work well even in this low-fluorescence regime. We therefore prefer to accept the approximation to zero which, as pointed out, is a minor effect.

L125-126: precision "If the subtraction caused the final signal to be negative due to noise, the resulting value was set to zero to avoid numerical instabilities that our ML model would not be able to deal with."

The authors are careful to sample fresh pollen quickly which is appropriate for a laboratory study. I believe, however, that there is reasonable evidence from recent measurements (see Hughes et al., ES&T Lett. 2020 for example) that pollen ruptures into smaller particles when exposed to humid conditions. It seems that the size and shape parameters are quite important for telling these 7 pollen types apart, could the authors comment on how this method might perform if there were pollen fragments in atmospheric measurements?

This is a very pertinent comment, as fragments seem to affect allergic people as much, if not more, as pollen does. Pollen fragments are typically very small compared to pollen grains; at most 2.5 μm while pollen grains usually have a minimal size of 10 μm. In that regard, pollen fragments would not pass the "pollen-non pollen" filter because of their size. If we were not using any filter, the holographic images of particles of size 2.5 μm or less would display a sort of dot, due to the low image resolution of 0.595 μm/pixel, with no characteristic shape. In that regard, the morphology of the particle would not be helpful to distinguish particles. However, we could count the number of particles of a certain size as an estimation of what could be pollen fragments. Besides, atmospheric conditions can also influence pollen grains' shape directly, but if the model is trained properly, it should cope with some variability in the data and still predict the correct class.

L272-273: added a sentence on the limitation regarding fragments, "For example, in ambient air, pollen can break into fragments also impacting allergy sufferers but not currently monitored."

Generally, I think the **limitations of the controlled nature of the study and the small number of homogeneous samples used, should be discussed more clearly throughout.** Although this offers a nice proof of concept that fluorescence may help discern different pollen types from one another, there is still the potential that a more diverse and variable ambient particles population will not be so readily separated. It would have been interesting, rather than focusing on the most allergenic species to, instead, focus on the most common species as a function of season as that would be more likely to tell you whether this method is capable of giving actionable information to allergy-prone citizens.

The most allergenic species that were selected for this study also comprise the most common species. *Betula* tree and *Cynosurus* grass (Poaceae) represent the most abundant pollen types in

central Europe with the heaviest impact in terms of allergies. The seasons of the seven taxa that were selected here cover the whole pollen season for central Europe, from December to July.

We emphasized the limitations of this study by modifying:

L259-261: "In addition, the main limitation of this study, focusing on a reduced number of pollen taxa and manually generated data, should be overcome in following work by gathering more data to train a broader model and test it on operational data."

Line 217 – I think you mean "averages" rather than accuracies

"Overall accuracies" here means the average accuracies of each model over the seven classes. We indeed mean accuracies but the global accuracy, not the specific accuracy of a model for one class.

For Figure 4, is that based on one of the two instruments used in the study? If so which one? How different would the discernment be if the other instrument was used?

Figure 4 is based on the two instruments; it displays all the data used in the study. We checked if there was a major difference between instruments by colouring the UMAP accordingly and also through the use of boxplots. In the UMAP case, for each taxon the colours of the two instruments were mixed up, showing that the pollen measured by both instruments gave similar measurements. The boxplots showed no major difference between the two instruments as quartiles clearly overlapped. Even if we tested this difference, it would not be possible to determine if the variability that we observe is due to the instruments themselves or due to natural variance in the samples.

We thank the Referee for their suggestions and remarks. We appreciate that you addressed specific issues in our paper and hope that we answered satisfactorily to improve the quality of the article.

---

## Author Response (AR2)

**Public justification (visible to the public if the article is accepted and published)**: I recommend publication with a technical correction. Lines 18, 236, 264, and 280 refer to "manually generated data". I think this would be more accurately described as "measurements of re-aerosolised pollen" or similar wording, because it was the pollen samples that were manually generated and not the data.

We thank the associate editor for their work and implication in improving our paper. Your recommendation regarding the "manually generated data" makes absolute sense and was corrected accordingly, underlining that the aerosolisation process was done by us but not the data itself. Thanks for noticing this.